# 3D-Printed Polyamide 12/Styrene–Acrylic Copolymer–Boron Nitride (PA12/SA–BN) Composite with Macro and Micro Double Anisotropic Thermally Conductive Structures

**DOI:** 10.3390/polym15132780

**Published:** 2023-06-22

**Authors:** Minhang Chen, Xiaojie Chen, Junle Zhang, Bingfeng Xue, Shangyu Zhai, Haibo She, Yuancheng Zhang, Zhe Cui, Peng Fu, Xinchang Pang, Minying Liu, Xiaomeng Zhang

**Affiliations:** 1School of Materials Science and Engineering, Henan Key Laboratory of Advanced Nylon Materials and Application, Engineering Laboratory of High-Performance Nylon Engineering Plastics of China Petroleum and Chemical Industry, Zhengzhou University, Zhengzhou 450000, Chinalmy@zzu.edu.cn (M.L.); 2The State Key Laboratory of Polymer Materials Engineering, Polymer Research Institute, Sichuan University, Chengdu 610065, China; 3Faculty of Engineering, Huanghe Science and Technology University, Zhengzhou 459000, China; 4Jinguan Electric Co., Ltd., Nanyang 473000, China

**Keywords:** polyamide, boron nitride, thermal conductivity, orientation, double anisotropic structure

## Abstract

Anisotropic thermally conductive composites are very critical for precise thermal management of electronic devices. In this work, in order to prepare a composite with significant anisotropic thermal conductivity, polyamide 12/styrene–acrylic copolymer–boron nitride (PA12/SA–BN) composites with macro and micro double anisotropic structures were fabricated successfully using 3D printing and micro-shear methods. The morphologies and thermally conductive properties of composites were systematically characterized via SEM, XRD, and the laser flash method. Experimental results indicate that the through-plane thermal conductivity of the composite is 4.2 W/(m·K) with only 21.4 wt% BN, which is five times higher than that of the composite with randomly oriented BN. Simulation results show that the macro-anisotropic structure of the composite (caused by the selective distribution of BN) as well as the micro-anisotropic structure (caused by the orientation structure of BN) both play critical roles in spreading heat along the specified direction. Therefore, as-obtained composites with double anisotropic structures possess great potential for the application inefficient and controllable thermal management in various fields.

## 1. Introduction

Thermally conductive composites composed of polymers and thermally conductive fillers are widely used as thermal management materials for electronic devices, due to low cost, light weight, and outstanding processability [1,2,3,4,5]. However, with the further development of electronic devices’ integration and high performance, heat generation increases significantly. As reported, if undesired heat cannot be dissipated effectively, the lifespan and running stability of electronic devices will be seriously reduced [6,7]. Therefore, improving the thermal management property of the composite applied as the packing material for devices is very critical and nonnegligible [8,9,10,11,12,13].

The construction of an orientation structure of anisotropic fillers is an effective method to take full advantage of fillers and improve the thermal management properties of composites [14,15,16,17,18,19]. For example, BN possesses a highly anisotropic thermally conductive property with an in-plane thermal conductivity of 600 W/(m·K) and through-plane thermal conductivity of only 2–30 W/(m·K) [20,21,22,23,24]. As a result, when BN is aligned along the in-plane direction, a composite with highly anisotropic thermal conductivity can be obtained. As reported by Guo, when the content of BN is 40 wt%, the thermal conductivity of the polyethylene-based composite along the in-plane direction can reach 3.6 W/(m·K), while the thermal conductivity along the perpendicular direction is only 0.7 W/(m·K) [16].

In addition to the micro-orientation structure of fillers, the macro-anisotropic structure of a composite also plays an important role in improving its anisotropic thermally conductive property and thermal management property [25,26]. For example, Wu and Fu et al. prepared PDMS/PEMF (polyethylene microfiber) and PDMS/RC-BNNS (regenerated cellulose-boron nitride nanosheet) composites with selectively distributed and vertically oriented PEMF or BNNS structure through the mold fixation method; results demonstrate that with the dual influences of the micro-orientation structure of fillers and the macro-anisotropic structure of the composite (caused by the selective distribution structure of fillers), the anisotropic thermally conductive property was significantly improved [27,28].

Inspired by the work mentioned above, and for further optimization of the design and preparation process, 3D printing coupled with the micro-shear method was applied in this work for the fabrication of the thermally conductive composite with macro and micro double anisotropic structures. In detail, as shown in Figure 1, through the macrostructure design via 3D printing and the subsequent filling process, fillers can be selectively distributed in the specified region, which can endow the composite with the macro-anisotropic structure and localized thermally conductive property [29]. Then, using the micro-shear method, fillers can be forced to align along the shear direction and form the micro-orientation structure. As a result, a thermally conductive composite with macro and micro double anisotropic structures can be obtained. The proposed method possesses the following merits: (1) Different from the existing methods, the distribution area of BN can be regulated freely and easily using 3D printing technology, which is of great significance for the controllable design and fabrication of thermally conductive composite or other composite parts, and properties of composites can be tuned via printing parameters [30,31,32]. (2) The micro-shear is generated through the assembly process of the two matching template materials with the assistance of external pressure. Compared with other orientation methods, the micro-shear method is much easier and not limited by the size and shape of the composite.

Therefore, based on this idea, a composite with macro and micro double anisotropic structures was prepared successfully. The effects of macro- and micro-anisotropic structures on thermal conductivity were investigated systematically. Furthermore, the heat dissipation behavior of such macro and micro double anisotropic structures was revealed in depth through finite element analysis (FEA). The study opens a new avenue for the design and fabrication of high-efficiency thermal management materials for electronic devices.

## 2. Experimental Section

### 2.1. Materials

Hexagonal BN (10–15 μm) was purchased from Dandong Rijin Technology Co., Ltd (Dandong, China). The PA12 (FS3300PA) for 3D printing was purchased from Hunan Farsoon High-Technology Corporation. Styrene–acrylic copolymer emulsion (SA) was obtained from Liaocheng Lugong Coating Additives CO., LTD (Liaocheng, China).

### 2.2. Specimen Preparation

The fabrication process of the composite is presented in Figure 1. (1) A series of matching orientation template material and matrix template material with the diameter of 25.4 mm are designed and fabricated via a 3D printing method, as shown in Appendix A. Parameters used in the 3D printing process are shown in Appendix A. (2) Fill SA–BN blend (SA:BN = 2:3) into the matrix template materials, forming underfill template composite. (3) Put the orientation template material right over the underfill template composite, then compress them together to force BN to orient along the thickness direction. (4) At last, after curing of SA–BN blend and polish the upper plane, the orientation template composites are obtained.

As shown in Figure 2, for comparison, the template composites and the isotropic composites are also prepared. Additionally, the preparation process and composition of these materials are shown in the Appendix A.

### 2.3. Characterizations

Dispersion and distribution of BN in composites were characterized using SEM (SU3500). Orientation degree of BN was calculated based on the results of XRD (SmartLab SE). Thermal diffusivities (α) were measured via the laser flash method (LFA467). Specific heat capacities (*C_p_*) were also measured via the laser flash method (LFA 467), and thermal conductivity was equal to the product of thermal diffusivity, specific heat capacity, and density (*ρ*). Macro heat dissipation processes of different composites were recorded using an infrared camera.

## 3. Results and Discussion

### 3.1. Macro- and Micro-Structures of Different Composites

First, macro-structures of the orientation template material, matrix template material, and the final orientation template composite are shown in Figure 2 and Appendix A. As shown in Appendix A, it can be seen that the shape and size of the template material can be accurately fabricated as designed through the 3D printing method. Moreover, the matching orientation template material and matrix template material can assemble together perfectly and easily, and the orientation template composite can be obtained. Then, the micro-structures of BN inside the orientation template composite were observed along different directions via SEM, as shown in Figure 3. The right views of SA–BN blend inside the template composite are presented in Figure 3b–d. BN is almost completely oriented along the shear plane, which is beneficial and critical for controllable and precise thermal management. Positive views of the SA–BN blend inside the orientation template composite demonstrate that BN particles closer to the shear plane possess higher orientation degrees, as shown in Figure 3e–g. Such morphology is consistent with the mechanism of the micro-shear field, and a continuous network has been formed in the orientation template composite. As shown in Figure 3h–j, BN is randomly oriented and distributed in the isotropic sample, and there is no filler network inside the composite. More details are shown in Appendix A.

As reported, the degrees of orientation for BN can be calculated using the ratio between I(002) and I(001) [33,34]. As shown in Figure 4 and Appendix A, compared with other composites, the orientation degree of BN in the orientation template composite is much higher. For example, when the filler content is 21.4 wt%, the ratio between the I(002) peak and I(100) peak of BN in the orientation template composite is almost four times as much as that in the isotropic sample. Therefore, it can be concluded that BN particles are selectively distributed in the orientation template composite, and the shear force generated during the assembling process is strong enough to align BN along the shear planes.

### 3.2. Thermal Conductivity and Enhancement

The effects of the macro-anisotropic structure of the composite (caused by the selective distribution of BN) and the micro-orientation structure of BN on the thermally conductive property of the composite are deeply investigated in this part. First, in order to evaluate the effect of the macro-anisotropic structure on thermal conductivity, the template composites without oriented BN were fabricated, as described in the experimental section. As shown in Figure 5a, the red line represents the thermal conductivities of template composites with various BN content, and the blue line corresponds to the thermal conductivities of the isotropic composite with various BN content. Results demonstrate that the template composites possess much higher through-plane thermal conductivities, due to the thermally conductive network formed in the composite. Therefore, it can be concluded that the macro-structure (selective distribution of BN) plays a critical role in enhancing the thermal conductivity of the composite.

Then, the effect of the micro-orientation structure of BN on thermal conductivity of the composite was investigated through the comparison between the orientation template composite and the template composite. The main difference between these two composites is the micro-orientation structure of BN, as shown in Figure 2. As shown in Figure 5a, the purple line represents the thermal conductivities of orientation template composites with various BN content, and the red line corresponds to the thermal conductivities of template composites. Results indicate that when the BN content is 21.4 wt%, the thermal conductivity of the orientation template composite is 4.2 W/(m·K) and 1.5 times that of the template composite. Detailed comparisons are shown in Figure 5b,c based on Equations (1) and (2), where β and γ are thermal conductivity enhancements and λ_1_, λ_2,_ and λ_3_ are thermal conductivities of the orientation template, template, and isotropic composites, respectively. Compared with the template and isotropic composites, the orientation template composite possesses more than 50% and 200% greater thermal conductivity.

Therefore, both the macro-anisotropic structure (caused by selective distribution of BN) and the micro-anisotropic structure (caused by orientation structure of BN) have significant effects on improving the thermally conductive property of the composite, and when combining them together, the thermal conductivity of the composite can be further increased, as shown in Figure 5e. Moreover, in comparison with other reported results, the orientation template composites also exhibit excellent thermally conductive properties, as shown in Figure 5d [35,36,37,38,39,40,41,42,43,44,45,46,47,48,49,50,51].
(1)$$\beta =\frac{\lambda_{1}{-\lambda }_{2}}{\lambda_{2}}$$
(2)$$\gamma =\frac{\lambda_{1}{-\lambda }_{3}}{\lambda_{3}}$$

### 3.3. Simulations for Thermal Conduction

Finite element analysis is applied to investigate the effects of macro and micro double anisotropic structures on thermal conduction of the composites. As shown in Figure 6, the filler structure inside the models is simplified based on the actual macro and micro morphologies of the composite. In the simulation, a heat source with the initial temperature of 120 °C is loaded at the middle of the lower surface of the model. Detailed simulation parameters are shown in Appendix A. Then, heat flux and temperature distribution of different models are calculated using Equations (3)–(5).
(3)$$\rho c\frac{𝜕T}{𝜕\tau }=\frac{𝜕}{𝜕x}\left(\lambda \frac{𝜕T}{𝜕x}\right)+\frac{𝜕}{𝜕y}\left(\lambda \frac{𝜕T}{𝜕y}\right)$$
(4)$$q_{\mathrm{xi}}=-\lambda \frac{𝜕T}{𝜕x}{\;q}_{\mathrm{yi}}=-\lambda \frac{𝜕T}{𝜕y}$$
(5)$$q_{\mathrm{total}}=\overset{n}{\underset{1}{\sum }}{(q}_{\mathrm{xi}}{+q}_{\mathrm{yi}})$$
where *q* is the heat flux, *λ* is the thermal conductivity, *τ* is the time, *c* is the specific heat capacity, *ρ* is the density, and *T* is the temperature.

First, as shown in Figure 6a, two different models with the same size and BN content were constructed to calculate the effect of the micro-orientation structure on thermally conductive behavior. The simulated results at 0.3 s are shown in Figure 6c; the orientation template composite possesses obviously higher thermal conduction due to the oriented BN near the boundary. Then, temperatures of the nodes along the different dot lines are extracted in sequence, as shown in Figure 6b. The comparison between temperatures of lines A and A′ can reflect vertical thermal conduction of two different models directly. As shown in Figure 6d with a black line and red line, it can be seen that temperatures of the nodes on line A are always higher than those of the nodes on line A′. Such results demonstrate that the orientation template composite model possesses much higher thermal conductivity along the vertical direction. Similarly, the temperatures of nodes on lines C and C′ are also compared, and results are shown in Figure 6e with blue and green lines. Near the heat source (bottom), temperatures of the nodes on line C are much lower than those of the nodes on line C′. Such results indicate that the oriented BN can dissipate heat along the vertical direction effectively and limit its spread. Furthermore, as shown in Figure 6d,e, the highly oriented structure of BN near the shear plane in the orientation template composite is beneficial for improving anisotropic and comprehensive thermal conductivity.

Similarly, the temperatures of nodes on lines B and B′ corresponding to transient heat source temperatures are extracted to evaluate overall thermal management properties of different models. As presented in Figure 6d, temperatures of the nodes on line B are always lower than those of the nodes on line B′, which clearly demonstrates that the oriented BN plays a critical role in enhancing comprehensive thermal conductivity. Furthermore, the sums of heat flux, which represent the comprehensive thermally conductive power, are also calculated and shown in Appendix A. Results further verify the conclusions. Then, the effect of the macro-anisotropic structure (caused by the selectively distributed structure of BN) on the thermally conductive behavior was investigated in this part via a comparison between the template composite model and isotropic composite model. As shown in Figure 7a, the temperatures of the nodes on line A′ are significantly higher than those of the nodes on line A″, yet the temperatures of the nodes on the line B′ are always lower than those of the nodes on the line B″. Such a phenomenon clearly demonstrates that the template composite model with the macro-anisotropic structure (selectively distributed structure of BN) possesses much higher through-plane and comprehensive thermal conductivity. Moreover, as presented in Figure 7b and Appendix A, the template composite model exhibits much lower temperature along the in-plane direction, which implies that the macro-anisotropic structure (selectively distributed structure of BN) plays a critical role in endowing the template composite with the unidirectional and localized thermally conductive property. In addition, as shown in Appendix A, when using a constant heat source, the trend and conclusion of simulation are not changed. Therefore, the simulated results clearly demonstrate that the orientation template composite with macro and micro double anisotropic structures possesses significant anisotropic thermal conduction behavior and can be effectively used as a precise and controllable thermal management material.

### 3.4. Infrared Thermal Images of Different Composites

With the guidance of the simulated results, the real-time heat dissipation processes of the composites were recorded using infrared thermal imaging technology. During the measurement, all of the comparative samples were put onto a plane heat source with the constant temperature of 80 °C. The digital images of the tested samples are presented in Figure 8a, and results of the heating and cooling processes are shown in Figure 8b. Notably, in comparison with the isotropic sample, the orientation template composite with the same BN content of 21.4 wt% possesses much faster thermal conduction along the through-plane direction. Furthermore, due to the great difference in thermal conductivity between the SA–BN part and PA12 part in the template composite and orientation template composite, heat mainly dissipates to the outside through the SA–BN part; thus, the orientation template composite shows an obvious localized thermally conductive characteristic. Figure 8c,d show the detailed temperature–time curves of different composites during the heating and cooling processes separately. In detail, after heating for 7 s, temperatures of the pure PA12, isotropic composite, template composite, and orientation template composite are 33, 43, 55, and 59 °C, respectively. Correspondingly, the orientation template composite also possesses much higher cooling efficiency. As a result, the orientation template composite not only exhibits highly efficient heat dissipation, but also has the characteristic of a localized thermally conductive property, which is very important for thermal management of complex electronic devices.

## 4. Conclusions

The PA12/SA–BN orientation template composite with macro and micro double anisotropic structures was fabricated successfully using the 3D printing and micro-shear methods. Results demonstrate that the BN near the shear plane is highly oriented, and 4.2 W/(m·K) of the through-plane thermal conductivity can be obtained with a BN content of only 21.4 wt%. The simulated results indicate that the macro-anisotropic structure (caused by the selective distribution of BN) and the micro-orientation structure of BN both play critical roles in enhancing thermal conductivity and selective heat dissipation properties. Therefore, the orientation template composite, with its efficient and controllable thermal management property, can effectively be used for packaging materials of complex electronic devices.

## Figures and Tables

**Figure 1 polymers-15-02780-f001:**
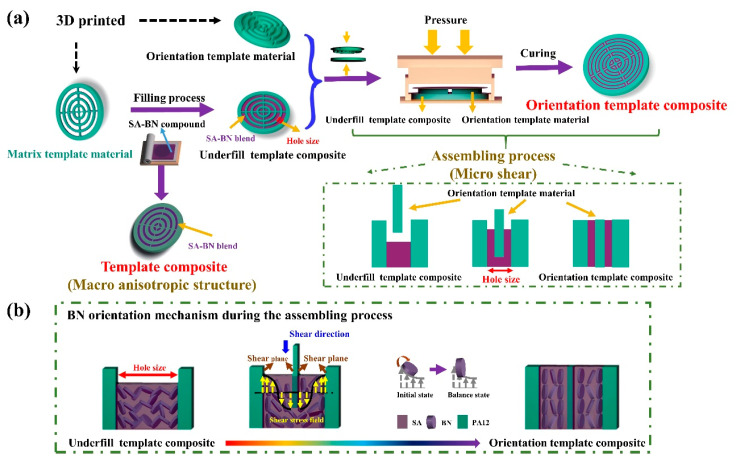
(**a**) Fabrication process of PA12/SA–BN orientation template compo site and (**b**) orientation mechanism of BN during the assembling process.

**Figure 2 polymers-15-02780-f002:**
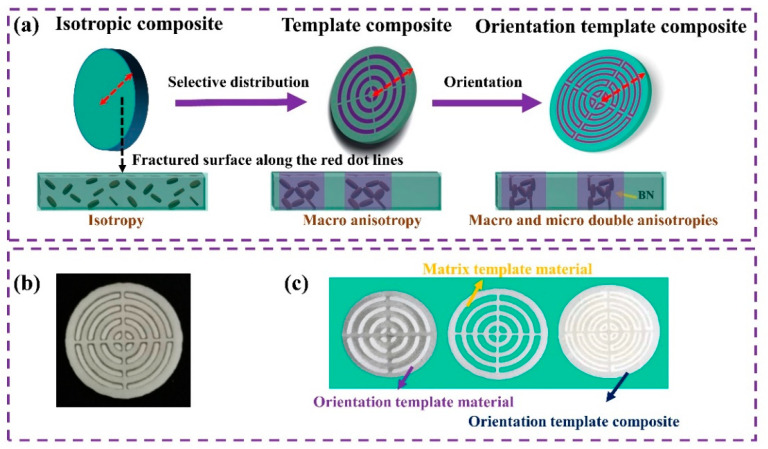
(**a**) The schematic images of different composites; (**b**) the images of the 3D-printed matrix template material and the corresponding orientation template material (assembled together); and (**c**) 3D-printed orientation template material, 3D-printed matrix template material, and the final orientation template composite (from left to right).

**Figure 3 polymers-15-02780-f003:**
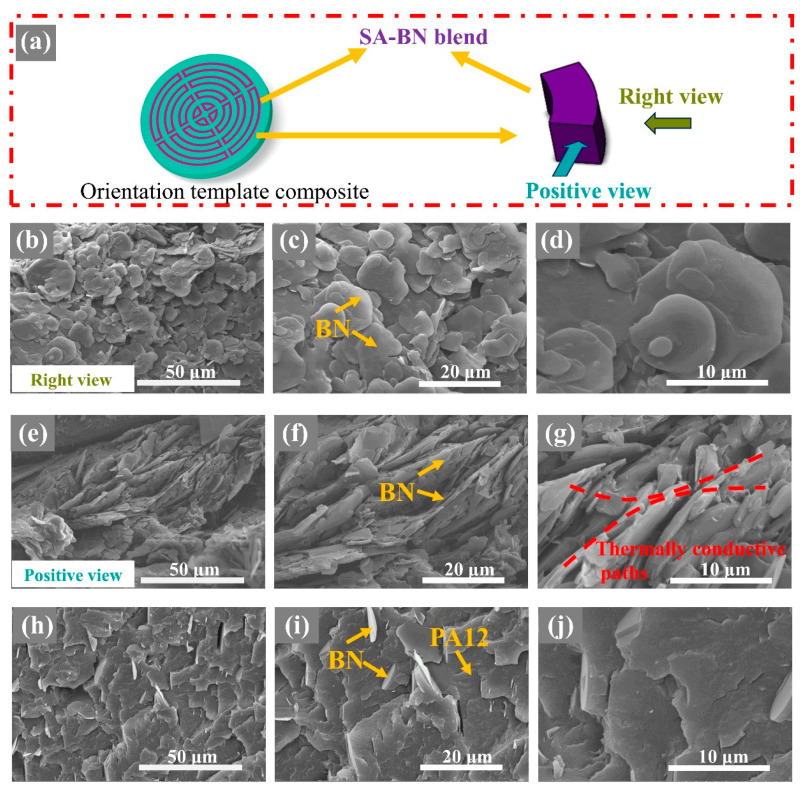
(**a**) The scheme of the observation direction and position during SEM measurement, (**b**–**d**) right views and (**e**–**g**) positive views of SA–BN blend inside the PA12/SA–BN orientation template composite, and (**h**–**j**) fractured surfaces of the PA12/BN isotropic composite. (7.8 wt% BN).

**Figure 4 polymers-15-02780-f004:**
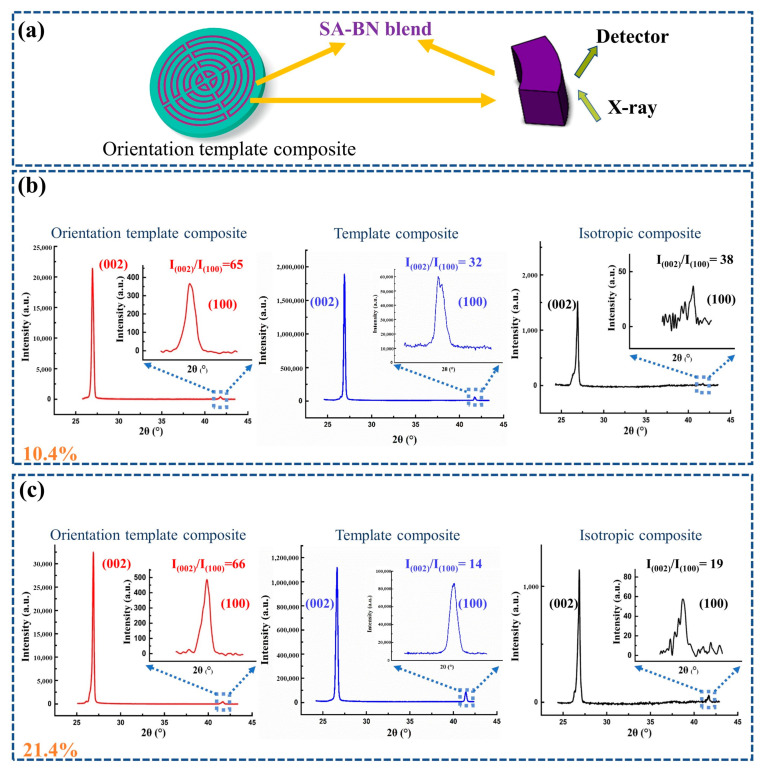
(**a**) The scheme of the testing direction during XRD measurement; XRD patterns of different composites with BN content of (**b**) 10.4 wt% and (**c**) 21.4 wt%.

**Figure 5 polymers-15-02780-f005:**
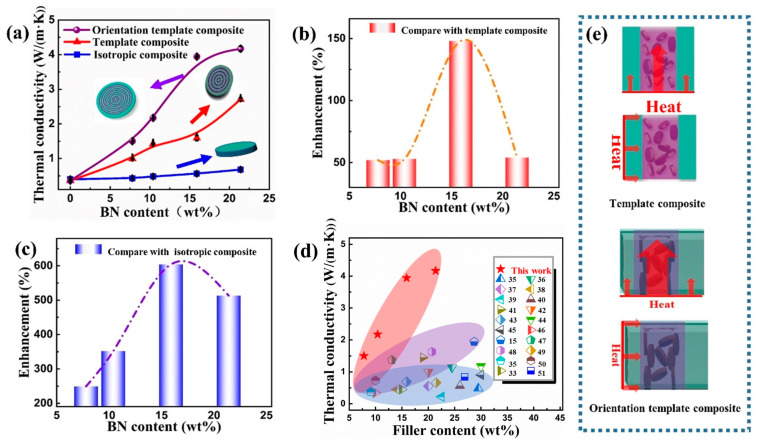
(**a**) Thermal conductivities of different composites along through-plane direction, thermal conductivity enhancements of the orientation template composite compared with (**b**) template composite and (**c**) isotropic composite, (**d**) comparison of thermal conductivities with other works, and (**e**) anisotropic thermal conduction mechanism of template and orientation template composites.

**Figure 6 polymers-15-02780-f006:**
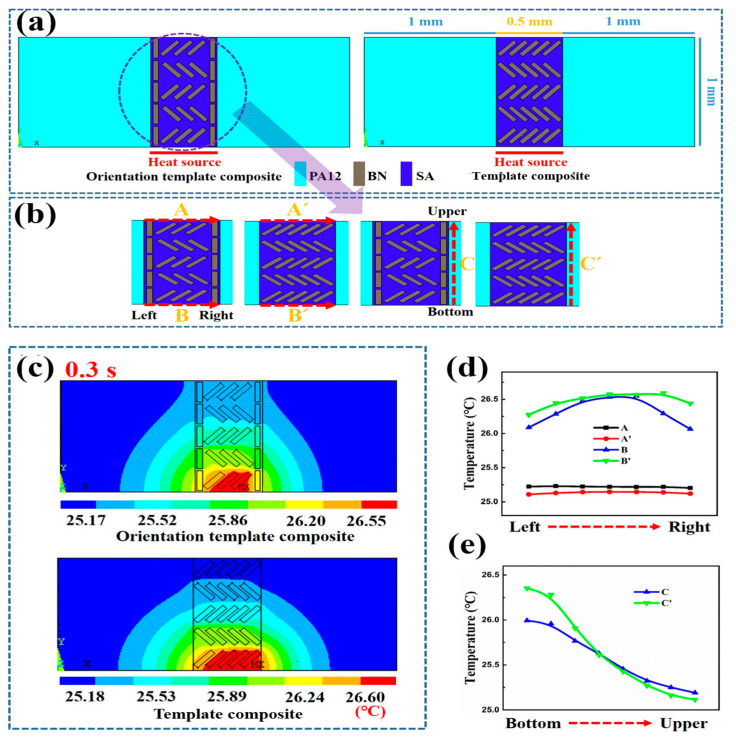
(**a**) The simplified models for simulations of template and orientation template composites, (**b**) locally magnified images of the model and the selected lines, (**c**) temperature contours, and (**d**,**e**) temperature–position curves in different models.

**Figure 7 polymers-15-02780-f007:**
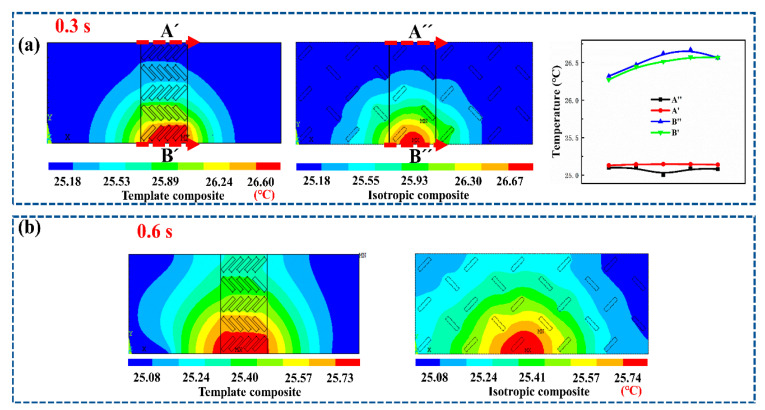
Temperature contours and temperatures of different lines at (**a**) 0.3 s and (**b**) 0.6 s.

**Figure 8 polymers-15-02780-f008:**
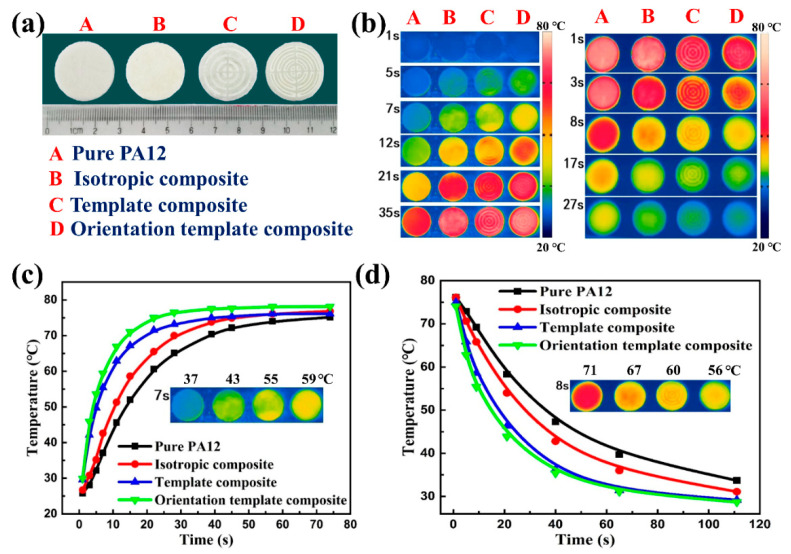
(**a**) Images of different samples; (**b**–**d**) heating and cooling processes of different composites recorded using an infrared thermal imager and corresponding temperature–time curves.

## Data Availability

The research data shown in the manuscript can be shared on request.

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
