# Peer review of "3D-Printed Polyamide 12/Styrene–Acrylic Copolymer–Boron Nitride (PA12/SA–BN) Composite with Macro and Micro Double Anisotropic Thermally Conductive Structures"

_polymers, 2023, doi:10.3390/polym15132780_

Round 1
Reviewer 1 Report
In this work, the polyamide 12/styrene-acrylic copolymer-boron nitride (PA12/SA-BN) composites with macro and micro double anisotropic structures are fabricated successfully using 3D printing and micro shear methods. The article is well-organized and written, and the images are of high quality. Also, the article's title is practical and attractive, but the following points should be considered before publishing.
The abstract could be written better and needs minor revisions. The purpose of research and innovation should be clearly stated. Also, the performed tests should be presented first, and then the results should be presented quantitatively and qualitatively. The article needs general writing and grammar editing.
The use of general sentences with more than four references can be seen in all parts of the introduction. For example, reference numbers 1-5, 8-13, 14-20, and 21-26 on the first page. On the other hand, appropriate references were not used to analyze the results. The introduction is written very briefly, and at the end, a suitable summary of the importance of the present issue is not provided.
The introduction needs to be reformed and deepened. Use the following new resources to complete this section. Statistical and experimental analysis of process parameters of 3D nylon printed parts by fused deposition modeling: response surface modeling and optimization. A New Strategy for Achieving Shape Memory Effects in 4D Printed Two-Layer Composite Structures. Development of Pure Poly Vinyl Chloride (PVC) with Excellent 3D Printability and Macro‐and Micro‐Structural Properties.
Figure 1 should be presented after the reference and the first mention in the text. Summarize section 2.2. Print parameters should be mentioned and added. How many thermal conductivity test samples were prepared for each group? How is the reproducibility of results checked? The accuracy of measuring equipment should be mentioned.
The results section is well organized and categorized. But some parts of it are just reporting the results. It is proposed to provide potential applications for the constructed structures. The advantage of the presented method compared to the conventional method (thermal conductive filament printing) should be stated, and it is better to make a comparison. It is suggested to modify the conclusion section as well as the abstract.
Ni comment.
Author Response
To the Reviewer 1
Reviewer 1: In this work, the polyamide 12/styrene-acrylic copolymer-boron nitride (PA12/SA-BN) composites with macro and micro double anisotropic structures are fabricated successfully using 3D printing and micro shear methods. The article is well-organized and written, and the images are of high quality. Also, the article's title is practical and attractive, but the following points should be considered before publishing. The abstract could be written better and needs minor revisions. The purpose of research and innovation should be clearly stated. Also, the performed tests should be presented first, and then the results should be presented quantitatively and qualitatively. The article needs general writing and grammar editing.
The use of general sentences with more than four references can be seen in all parts of the introduction. For example, reference numbers 1-5, 8-13, 14-20, and 21-26 on the first page. On the other hand, appropriate references were not used to analyze the results. The introduction is written very briefly, and at the end, a suitable summary of the importance of the present issue is not provided. The introduction needs to be reformed and deepened. Use the following new resources to complete this section. Statistical and experimental analysis of process parameters of 3D nylon printed parts by fused deposition modeling: response surface modeling and optimization. A New Strategy for Achieving Shape Memory Effects in 4D Printed Two-Layer Composite Structures. Development of Pure Poly Vinyl Chloride (PVC) with Excellent 3D Printability and Macro‐and Micro‐Structural Properties.
Figure 1 should be presented after the reference and the first mention in the text. Summarize section 2.2. Print parameters should be mentioned and added. How many thermal conductivity test samples were prepared for each group? How is the reproducibility of results checked? The accuracy of measuring equipment should be mentioned. The results section is well organized and categorized. But some parts of it are just reporting the results. It is proposed to provide potential applications for the constructed structures. The advantage of the presented method compared to the conventional method (thermal conductive filament printing) should be stated, and it is better to make a comparison. It is suggested to modify the conclusion section as well as the abstract.
First of all, we acknowledge your comments and suggestions very much, which are valuable in improving the quality of our manuscript. Hopefully, we have addressed all of your concerns. We revised our manuscript in accordance with your instructive guidance and we feel that the revised manuscript is a great improvement on the original.
- The abstract could be written better and needs minor revisions. The purpose of research and innovation should be clearly stated. Also, the performed tests should be presented first, and then the results should be presented quantitatively and qualitatively. The article needs general writing and grammar editing.
Response: Thanks for your professional suggestion and reminding, the abstract was revised as suggested, and writing problems were also solved.
- The use of general sentences with more than four references can be seen in all parts of the introduction. For example, reference numbers 1-5, 8-13, 14-20, and 21-26 on the first page. On the other hand, appropriate references were not used to analyze the results. The introduction is written very briefly, and at the end, a suitable summary of the importance of the present issue is not provided. The introduction needs to be reformed and deepened. Use the following new resources to complete this section. Statistical and experimental analysis of process parameters of 3D nylon printed parts by fused deposition modeling: response surface modeling and optimization. A New Strategy for Achieving Shape Memory Effects in 4D Printed Two-Layer Composite Structures. Development of Pure Poly Vinyl Chloride (PVC) with Excellent 3D Printability and Macro‐and Micro‐Structural Properties.
Response: Thanks for your professional suggestion and reminding, the problems about references and introduction sections were revised. And recommended references were cited, they are very helpful for improving quality of our manuscript.
- Figure 1 should be presented after the reference and the first mention in the text. Summarize section 2.2. Print parameters should be mentioned and added. How many thermal conductivity test samples were prepared for each group? How is the reproducibility of results checked? The accuracy of measuring equipment should be mentioned. The results section is well organized and categorized. But some parts of it are just reporting the results. It is proposed to provide potential applications for the constructed structures. The advantage of the presented method compared to the conventional method (thermal conductive filament printing) should be stated, and it is better to make a comparison. It is suggested to modify the conclusion section as well as the abstract.
Response: Thanks for your reminding. The position of Figure 1 was revised. The specific parameters about 3D printing have been supplemented in the supplementary file, as shown in Table R1. And conclusion section was also been revised.
Table R1. Parameters about 3D printing.
|
Parameters |
Value |
|
Powder supply cylinder temperature |
150℃ |
|
Molding cylinder temperature |
171.5℃ |
|
Laser power |
30 W |
|
Scan speed |
2.28 m/s |
|
Scan distance |
0.08 mm |
|
Layer thickness |
0.1 mm |
Thermal conductivity test: In the thermal conductivity test, each group contains three samples, and each sample was tested for 3 times. Reproducibility of results is very good, and the error is less than 5% for each sample.
Potential applications for the constructed structures: The orientation template composite with macro and micro double anisotropic structures possesses outstanding localized and unidirectional thermally conductive property, which makes it have unique advantages over other composites with uniform or isotropic thermally conductive property. For example, in the electronic devices, heat mainly generated by the high-power component, when using uniform or isotropic thermal management materials, heat will inevitably spread around and affect other low-power or no-power components. But by using the orientation template composite as thermal management material, heat can be dissipated from high-power component directly without affecting other components.
Method comparison: Filament printing is a general method to prepare materials with complex structure, and it is also suitable for preparing orientation template composites. Generally, materials prepared by filament printing possesses dense structure, but materials prepared by SLS always have porous structure. In the orientation template composites, porous matrix with thermal insulating property is better to further improve localized and unidirectional thermally conductive property.

Reviewer 2 Report
The work herein, the manuscript entitled “3D Printed Polyamide 12/Styrene-acrylic copolymer Boron Nitride (PA12/SA-BN) Composite with Macro and Micro Double Anisotropic Thermally Conductive Structures”, the work described in the manuscript is about the preparation of polyamide 12/styrene-acrylic copolymer - boron nitride (PA12/SA-BN) composites with macro and micro double anisotropic structures by using 3D printing and micro shear methods. The work provides important guidance for designing and fabricating high-efficiency anisotropic thermally conductive composite for the thermal management of electronic devices.
1. The novelty is not so high because, a part and many similarities of the present work is already published in „Composites Part B 225 (2021) 109267, Construction and mechanism of 3D printed polyamide 12/boron nitride template composites with localized and unidirectional thermally conductive property Minhang Chen a, Tingting Yin a, Peng Fu a, Haibo She c, Xiaomeng Zhang a,b,c,*, Wei Zhao a, Zhe Cui a, Xinchang Pang a, Qingxiang Zhao a, Minying Liu”.
1. In the SEM images (figure 3), the authors indicate where the boron nitride (BN) and PA12 appear. They are claiming that they performed in the same position as the XRD spectroscopy, but it is hard to believe. Therefore, I think it would be a good idea for the authors to measure together with SEM microscopy the EDX spectroscopy. In this way, they can determine the ratio between boron and carbon atom. The differences in the distribution of BN in the composite's macro- and micro- anisotropic structure can also be determined.
2. In the description, in the „2. Experimental Section”, the particle size of the BN is mentioned to be between 10-15 μm. It will be better if they include in Figure 3 also an SEM image of pure BN without polymer around. In this way, one can see how pure BN’s morphology looks.
3. To determine the content of BN in the copolymer matrix, the authors can analyse the prepared samples by FTIR spectroscopy using the analytical mode by measuring the same amount of the composite samples in the KBr pellet. BN has a distinctive adsorption band in the FTIR spectra, which can be deconvoluted and integrated.
By introducing the above, the present manuscript may look different in approach compared to the already published article in „Composites Part B 225 (2021) 109267”.
The English language needs some mino grammar corrections.
Author Response
To the Reviewer 2
Reviewer 2: The work herein, the manuscript entitled “3D Printed Polyamide 12/Styrene-acrylic copolymer Boron Nitride (PA12/SA-BN) Composite with Macro and Micro Double Anisotropic Thermally Conductive Structures”, the work described in the manuscript is about the preparation of polyamide 12/styrene-acrylic copolymer - boron nitride (PA12/SA-BN) composites with macro and micro double anisotropic structures by using 3D printing and micro shear methods. The work provides important guidance for designing and fabricating high-efficiency anisotropic thermally conductive composite for the thermal management of electronic devices.
First of all, we acknowledge your comments and suggestions very much, which are valuable in improving the quality of our manuscript. Hopefully, we have addressed all of your concerns. We revised our manuscript in accordance with your instructive guidance and we feel that the revised manuscript is a great improvement on the original.
- The novelty is not so high because, a part and many similarities of the present work is already published in, Composites Part B 225 (2021) 109267, Construction and mechanism of 3D printed polyamide 12/boron nitride template composites with localized and unidirectional thermally conductive property Minhang Chen a, Tingting Yin a, Peng Fu a, Haibo She c, Xiaomeng Zhang a,b,c,*, Wei Zhao a, Zhe Cui a, Xinchang Pang a, Qingxiang Zhao a, Minying Liu”.
Response: This work is an extension of the previous work. In the previous work, only macro anisotropic structure was obtained, but in this work, macro and micro double anisotropic structures by using new methods of 3D printing coupled with micro shear. Correspondingly, thermal conductivity and thermal management property were further enhanced.
- In the SEM images (figure 3), the authors indicate where the boron nitride (BN) and PA12 appear. They are claiming that they performed in the same position as the XRD spectroscopy, but it is hard to believe. Therefore, I think it would be a good idea for the authors to measure together with SEM microscopy the EDX spectroscopy. In this way, they can determine the ratio between boron and carbon atom. The differences in the distribution of BN in the composite's macro- and micro- anisotropic structure can also be determined.
Response: Thanks for your professional suggestion. In SA-BN compound, some of BN is covered by SA, as shown in Figure R1. Therefore, it is difficult to distinguish BN and SA or calculate their ratio by EDS spectroscopy, as shown in Figure R1(c).
Actually, BN can be easily distinguished by its special two-dimensional structure. As shown in Figure R1(a, b) and Figure R2, BN can be clearly observed.
Figure R1. (a) Pure SA, (b) SA-BN compound, and (c) EDS mapping image of SA-BN compound.
Figure R2. SEM images of (a) PA12 and (a) isotropic composite with the BN content of 33.5 wt%.
- In the description, in the, 2. Experimental Section”, the particle size of the BN is mentioned to be between 10-15 μm. It will be better if they include in Figure 3 also an SEM image of pure BN without polymer around. In this way, one can see how pure BN’s morphology looks.
Response: Thank you for your valuable comment. As shown in Figure R1(b), the observed platelets with smooth surfaces are corresponding to BN.
Figure R3. Morphology of pure BN.
- To determine the content of BN in the copolymer matrix, the authors can analyse the prepared samples by FTIR spectroscopy using the analytical mode by measuring the same amount of the composite samples in the KBr pellet. BN has a distinctive adsorption band in the FTIR spectra, which can be deconvoluted and integrated.
Response: Thank you for your valuable comment. The composite as-obtained in this work is a heterogeneous composite, which is difficult in the preparation of FTIR test sample. In addition, as is shown in Figure R1, due to BN is covered by SA, the accuracy of infrared results may be affected.

Reviewer 3 Report
In this work, authors fabricated polyamide 12/styrene-acrylic copolymer-boron nitride (PA12/SA-BN) composites using 3D printing and micro shear methods. They reported high through-plane thermal conductivity of the composite of 4.2 W/(m·K) with only 21.4 wt% BN, which is really impressive. The study is well conducted and the obtained results are worthy of publication and such that I can recommend the publication of this manuscript provided that the authors address the following comments:
1- One of the most important points for the thermal transport in composite materials is related the interfacial thermal resistance between the fillers and the matrix, so called kapitsa effect, which becomes critical and can dominate the heat transport in the nanocomposite and 2D heterostructures, this effect is not also included in this work. Although this effect is not included, authors may include some discussion related to this in the finite element modeling using the available data in the literature.
2- Please include more discussions concerning the size of the systems and applied boundary conditions, are the results insensitive to the sample size?
3- I recommend a general proofreading and cleanup of the language and grammar.
Author Response
To the Reviewer 3
Reviewer 3: In this work, authors fabricated polyamide 12/styrene-acrylic copolymer-boron nitride (PA12/SA-BN) composites using 3D printing and micro shear methods. They reported high through-plane thermal conductivity of the composite of 4.2 W/(m·K) with only 21.4 wt% BN, which is really impressive. The study is well conducted and the obtained results are worthy of publication and such that I can recommend the publication of this manuscript provided that the authors address the following comments: 1 One of the most important points for the thermal transport in composite materials is related the interfacial thermal resistance between the fillers and the matrix, so called kapitsa effect, which becomes critical and can dominate the heat transport in the nanocomposite and 2D heterostructures, this effect is not also included in this work. Although this effect is not included, authors may include some discussion related to this in the finite element modeling using the available data in the literature. 2 Please include more discussions concerning the size of the systems and applied boundary conditions, are the results insensitive to the sample size? 3 recommend a general proofreading and cleanup of the language and grammar.
First of all, we acknowledge your comments and suggestions very much, which are valuable in improving the quality of our manuscript. Hopefully, we have addressed all of your concerns. We revised our manuscript in accordance with your instructive guidance and we feel that the revised manuscript is a great improvement on the original.
- One of the most important points for the thermal transport in composite materials is related the interfacial thermal resistance between the fillers and the matrix, so called kapitsa effect, which becomes critical and can dominate the heat transport in the nanocomposite and 2D heterostructures, this effect is not also included in this work. Although this effect is not included, authors may include some discussion related to this in the finite element modeling using the available data in the literature.
Response: Thank you for your valuable suggestion. Kapitza thermal resistance has a serious effect on thermal conductivity, but kapitza thermal resistance mainly measured by TDTR technology or calculated by MD simulation. In this work, by using LFA method to measure thermal conductivity and finite element method to calculate thermal conductivity, kapitza thermal resistance cannot be obtained directly. Therefore, the effect of kapitza was not considered in this work.
In our following work, the effect of kapitza on macro thermally conductive property will be investigated in-depth.
- Please include more discussions concerning the size of the systems and applied boundary conditions, are the results insensitive to the sample size?
Response: Thank you for your valuable suggestions. The size of the systems as well as applied boundary conditions will affect the heat dissipation behaviors, but will not affect variation trend and corresponding conclusions. As shown in Figure R1-R3, changing heat sources, size of system and so on, models with macro and micro anisotropic structures always possess much better thermally conductive property.
Figure R1. The temperature of different composites under 70 ℃, 100 ℃, and 130 ℃ heats source respectively.
Figure R2. Sum of heat flux in different composites under 70 ℃, 100 ℃, and 130 ℃ heats source respectively.
Figure R3. The temperature contours of different systems with sizes of (a) 2.5◊1 mm and (b) 1.5◊1 mm under 100 ℃ heat source.
- Recommend a general proofreading and cleanup of the language and grammar.
Response: Thanks for your professional suggestion and reminding, the English editing problems were revised in the revised manuscript.

Round 2
Reviewer 2 Report
I agree with the publication of the manuscript in its present form.
Reviewer 3 Report
I recommend the publication